# Functional Interaction of Hypoxia-Inducible Factor 2-Alpha and Autophagy Mediates Drug Resistance in Colon Cancer Cells

**DOI:** 10.3390/cancers11060755

**Published:** 2019-05-30

**Authors:** Abril Saint-Martin, Jacobo Martínez-Ríos, M. Cristina Castañeda-Patlán, Miguel Angel Sarabia-Sánchez, Nydia Tejeda-Muñoz, Alberto Chinney-Herrera, Gloria Soldevila, Roberto Benelli, Paula Santoyo-Ramos, Alessandro Poggi, Martha Robles-Flores

**Affiliations:** 1Department of Biochemistry, Facultad de Medicina, Universidad Nacional Autónoma de México (UNAM), Mexico City 04510, Mexico; abrilsaint@hotmail.com (A.S.-M.); cristi_ccp@yahoo.com (M.C.C.-P.); mike_sarabia@hotmail.com (M.A.S.-S.); nydia.tejeda@gmail.com (N.T.-M.); paula_ab@yahoo.com (P.S.-R.); 2Department of Immunology, Instituto de Investigaciones Biomédicas, Universidad Nacional Autónoma de México (UNAM), Mexico City 04510, Mexico; jacobomartinezrios@yahoo.com.mx (J.M.-R.); alder_30@hotmail.com (A.C.-H.); soldevi@unam.mx (G.S.); 3Immunology unit, IRCCS Ospedale Policlinico San Martino, 16132 Genoa, Italy; roberto.benelli@hsanmartino.it; 4Molecular Oncology and Angiogenesis Unit, IRCCS Ospedale Policlinico San Martino, 16132 Genoa, Italy; alessandro.poggi@hsanmartino.it

**Keywords:** colon cancer, hypoxia-inducible factors, drug resistance, autophagy

## Abstract

Hypoxia and the accumulation of hypoxia-inducible factors (HIFs) in tumors have been associated with therapeutic resistance and with autophagy establishment. We examined the effects of stable knockdown of HIF-1α or HIF-2α expression on autophagy and drug resistance in colon cancer cells. We found that under normoxic conditions, malignant cells exhibit increased basal levels of autophagy, compared with non-malignant cells, in addition to the previously reported coexpression of HIF-1α and HIF-2α. Knockdown of HIF-1α or HIF-2α expression resulted in increased autophagic and apoptotic cell death, indicating that the survival of cells is HIF-dependent. Cytotoxic-induced cell death was significantly increased by knockdown of HIFs but not by autophagy inhibition. Strikingly, although malignancy-resistant cells were sensitized to death by nutrient stress, the combination with HIF-2α depletion, but not with HIF-1α depletion, induced severe cell death. Oxidative stress levels were significantly increased as a result of HIF-2α specific inhibition or silencing suggesting that this may contribute to sensitize cells to death. The in vitro results were confirmed in vivo using a xenograft mouse model. We found that coordinated autophagy and mTOR inhibition enhanced cell death and induced tumor remission only in HIF-2α-silenced cells. Finally, using a specific HIF-2α inhibitor alone or in combination with drugs in patient-derived primary colon cancer cells, overcame their resistance to 5-FU or CCI-779, thus emphasizing the crucial role played by HIF-2α in promoting resistance and cell survival.

## 1. Introduction

Hypoxia is a condition frequently found in the tumor microenvironment. The expression of hypoxia-inducible factors (HIFs) in tumor cells is crucial for the adaptation response to oxygen deprivation inducing angiogenesis, erythropoiesis, apoptosis, proliferation, and to favor tumorigenesis [1,2,3]. In this regard, it has been shown that HIFs participate in many aspects of tumor progression promoting metabolic reprogramming, neovascularization, apoptosis and autophagy induction, invasion, metastasis, and also drug resistance resulting in a poor prognosis for patients [4].

To date, three HIF-α isoforms have been identified in mammals that are encoded by different genes. However, whereas it was clearly demonstrated that HIF-1α and HIF-2α are key regulators of the hypoxia response and of tumorigenesis, the roles played by HIF3α, either in hypoxia or in cancer promotion are far less clear [4]. HIF-1α and HIF-2α display a high degree of sequence identity and are both regulated similarly by hypoxia. However, although they activate the transcription of some shared target genes, they are non-redundant and also activate unique target genes. Consistent with this, HIF-1α and HIF-2α deficiency have revealed isoform-specific effects on cancer cells, indicating that they can act in different ways to promote malignancy and cancer progression [3,4]. In agreement with this notion, we have reported that HIF-1α and HIF-2α play opposing roles in canonical Wnt signaling activation in colon cancer cells despite both being essential for stemness and malignancy maintenance [5].

It is important to take into consideration that cancer cells are characterized by high levels of HIFs expression also under normoxic conditions. This is due as a result of the high oncogenic signaling activation in cancer cells that induce O_2_-independent HIF-α accumulation by increased transcription and/or translation of HIF-α mRNA [3,6]. In this respect, we have also reported that under normoxic conditions colon cancer cells coexpress HIF-1α and HIF-2α, compared to non-malignant colon cells, which do not express these factors under the same conditions [5].

The expression of both HIFs, particularly of HIF-1α, has been associated with radio- and chemotherapy resistance [2,7,8,9]. Thus, disruption of HIF actions represents a great promise in anticancer therapy. On the other hand, the high levels of HIFs expression have been associated with increased autophagy induction in several cancer cell types. In this regard, it has been shown that cytotoxic treatment failure mediated by hypoxia can be produced by HIF-1α-dependent induction of autophagy [10]. However, the importance of the relationship between HIFs and autophagy in drug resistance establishment remains obscure. The PI3K/AKT/mTOR pathway is dysregulated in a high number of cancer types such as colon cancer and, thus, its pharmacological inhibition has been considered as an attractive approach for treatment. Among agents that interfere with this signaling pathway, several inhibitors of mTOR, such as CCI-779, which is an analog of rapamycin, have demonstrated efficacy in several cancer types [11].

In this study, we examined the effects of stable HIF-1α or HIF-2α siRNA knockdown on autophagy and drug resistance displayed by RAS-driven and BRAF-driven human colon carcinoma cell lines and in patient-derived primary colon cancer cells. We found that HIF-2α plays a crucial role in survival and resistance promotion. Our results show that autophagy mostly acts as a survival mechanism and that depletion of HIF-2α expression, either alone or in combination with autophagy and mTOR inhibition, greatly enhances cell death and induces tumor remission, demonstrating the reliance upon HIF-2α expression for survival.

## 2. Results

### 2.1. Human Colon Malignant Cells Show High Levels of Basal Autophagy and Knockdown of Hypoxia-Inducible Factors (HIFs) Expression Increases These Levels under Normoxic Conditions

To examine the basal levels of autophagy in colorectal cancer (CRC) cell lines, we selected the human RKO cell line and primary SW480 and its derivative metastatic SW620 cell lines as representative of a range of BRAF-driven and KRAS-driven colorectal genotypes, respectively [12]. Once autophagy is induced, LC3-I protein is converted to LC3II by proteolytical cleavage and lipidation, localizing LC3II to the autophagosome membrane. Therefore, it is a measure of autophagosome formation that can be examined either by Western blot or by confocal microscopy to visualize the appearance of punctate structures. As shown in Figure 1A, all three cancer cell lines, regardless of their genotype, showed increased levels of autophagy compared with 112CoN non-malignant cells, since they displayed higher ratios of detectable LC3-II/LC3-I bands on Western blots. To visualize the autophagosome puncta, the cells were transfected with the autophagy reporter enhanced green fluorescent protein (EGFP)-LC3 plasmid. The non-malignant colon cell line 112CoN was also transfected as a control. As can be seen in Figure 1B, all malignant cells showed higher basal levels of autophagosome puncta compared with nonmalignant cells. Consistent with this, the expression of other typical autophagy markers such as Beclin 1, Spag5, Atg7, Atg12 and Atg5, appeared augmented in colon cancer cells, compared with nonmalignant cells, as can be observed in Appendix A.

We have previously reported that HIF-1α and HIF-2α are co-expressed in colon cancer cells but not in nonmalignant 112CoN cells under normoxic conditions [5]. Consistent with this, the analysis of the expression of these factors by Western blotting, as shown in Figure 1C indicated that in the absence of hypoxia, HIF-1α and HIF-2α are expressed only in cancer cells. To analyze HIF function in autophagy-induced cell death, we created a stable knockdown of HIF-1α or of HIF-2α in SW480 malignant cells, which exhibit high basal autophagy levels under normoxic conditions. Stable transfected SW480 cells with the control scrambled shRNA plasmid, with HIF-1α RNAi, or with HIF-2α RNAi were selected by FACS on the basis of the silencing efficiency observed in comparison with control cells (higher than 70%). Selected transfectant clones were then analyzed by Western blotting as shown in Figure 1D. Consistent with previous reports, Figure 1D shows how the loss of one HIF-α subunit was compensated by upregulating the remaining HIFα isoform. Interestingly, we were unable to produce the simultaneous knockdown of both HIF factors in either RKO or SW480 cells, since all of them finally died (Appendix A), reinforcing the importance of these factors in survival promotion in colon cancer cells.

Next we examined the effect of the knockdown of HIF-1α or HIF-2α on autophagy levels in colon cancer cells. Control or HIF-silenced cells were transfected with the autophagy reporter-enhanced green fluorescent protein (EGFP)-LC3 plasmid. Cells were then grown in normal media on glass-bottom Petri dishes and examined by confocal microscopy for the presence of autophagosome puncta. Interestingly and in agreement with previous reports [12,13], knockdown of HIFs increased autophagy levels even more, particularly as a result of HIF-1α depletion, as visualized by increased punctate structures (Figure 1E) and by the conversion of LC3-I to LC3-II, shown in the Western blot and its corresponding bar graph (Figure 1F), which was also higher in HIFs-depleted cells than in control HIFs-expressing cells. Since it is well established that both HIF-α stimulate autophagy, we investigated whether this paradoxical increase in autophagy induction may be the result of the compensatory expression we previously observed when one HIF-α subunit is depleted. Because we could not get the simultaneous knockdown of HIF-1/2α factors, we made use of PT-2385 [14], a ligand that blocks specifically the heterodimerization of HIF-2α with HIF-1β needed to activate the transcription of genes regulated by HIF-2α to combine with HIF-silenced cells. The autophagy levels quantified by flow cytometry presented in Figure 1G show how the incubation of stable HIF-1α-silenced cells with the HIF-2α-specific antagonist reverted (indicated with a red arrow) the increase in autophagy levels produced as result of HIF-1α-silencing, which displayed a compensatory HIF-2α expression (Figure 1D). However, because in this type of cancer cells, HIF-independent autophagy is particularly robust, we cannot rule out that the stressful condition imposed by the depletion of one HIF-α isoform could also activate the HIF-independent autophagic pathway and thus the transcription and/or translation of HIFs induced by oncogenic activity, contributing to autophagy induction. In support of this, it is interesting to note in Appendix A that when we tried to transiently knockdown HIF-1α in stable silenced HIF-2α SW480 cells, surprisingly, the HIF-2α expression was recuperated after 24 h post-transfection with the sh-HIF-1α plasmid (indicated by a red arrow in the Figure).

### 2.2. Colon Cancer Cells Can Be Sensitized to Drug Treatment with Starvation but Combining These Conditions with HIF-2α Silencing Induced Severe Effects on Cell Death

Autophagy has been recognized as an important regulator of cellular viability under stressful conditions. Consistent with this, it has been reported that autophagy is induced as a mechanism by which cells may survive to cytotoxic drug treatment [10,15,16,17,18,19]. To examine the role of HIFs expression in cytotoxicity-induced cell death, we first evaluated the sensitivity of SW480 and RKO colon carcinoma cells to the chemotherapeutic drug 5-fluorouracil (5-FU), to the mTOR inhibitor CCI-779 and to the autophagy flux inhibitor HCQ. It is well known that the inhibition of mTOR, which is an essential regulator of autophagy, can mimic starvation or recapitulate aspects of hypoxia [20,21]. Autophagy may be abolished to increase the cytotoxicity of mTOR inhibition through HCQ, which prevents the fusion of autophagosomes with lysosomes, the final step of autophagy [11]. The cytotoxicity of each drug was evaluated by the MTT viability assay. As shown in Appendix A (SW480 cells) and S3 (RKO cells), HCQ did not affect cell viability at any of the concentrations tested after 48 h of treatment (or 24 h and 72 h), compared with untreated control cells. As has been reported, SW480 cells displayed resistance to 5-FU [22], since doses lower than 100 μM, did not significantly inhibit cell viability, even after 72 h of treatment, but at concentrations over 200 μM, inhibited the viability of SW480 cells in a time- and dose-dependent manner (Appendix A). Interestingly, the resistance displayed by RKO colon cancer cells against 5-FU or CCI-779 treatment was the opposite of that observed for SW480 cells (Appendix A), since RKO cells displayed resistance to CCI-779 treatment but were 5-FU sensitive, as can be observed in Appendix A. Based on these results, we chose the following doses of drugs for the next experiments: 20 μM of HCQ, which had no inhibitory effect on cell viability and was reported by other authors as a dose necessary to inhibit autophagy without affecting cell viability [11]; 300 μM of 5-FU, which had an approximately 50% inhibitory effect on SW480 cell viability; and 10 μM of CCI-779 (no cytotoxic effect) compared with 20 μM (cytotoxic effect).

To explore whether blocking of HIFs expression can sensitize malignant resistant cells to drug treatment, we investigated the effect on cell viability induced by 48 h of autophagy and/or mTOR inhibition (using HCQ and CCI-779 alone or in combination) in control or HIF-silenced SW480 cells. As shown in Figure 2A, although CCI-779 treatment alone induced SW480 cell death in control HIFs-expressing cells, it induced a significantly higher decrease in viability in HIF-silenced cells, and in general, cell viability mainly decreased as a result of HIF-1α or HIF-2α silencing. Remarkably, autophagy flux inhibition did not sensitize cells to death when combined with CCI-779 in HIF-silenced cells and even appeared to protect control HIFs-expressing cells from death when combined with CCI-779 (Figure 2A, fourth column).

To assess the effect of nutritional stress (starvation) on autophagy and/or apoptosis induction in this setting of HIFs expression, growth medium from SW480 control or HIF-silenced cells was replaced with Hanks’ balanced salt solution (HBSS) for 4, 8, 16, or 24 h. Control, HIF-1α or HIF-2α -silenced cells were then collected at these time points and assessed for autophagy by Western blotting to detect LC3-I conversion to LC3-II, and for apoptosis detecting the expression of cleaved caspase-3. The results shown in Figure 2B indicate that, as expected, nutritional stress-induced autophagy in both HIF-expressing (Control) or HIF-silenced SW480 cells, but both autophagy and apoptosis rates were higher in HIF-depleted cells than in the controls. In agreement with our previous report [5], we found here that the apoptosis rate was increased in colon cancer cells as a result of HIF-1α or HIF-2α knockdown (please see the corresponding 0 times in the apoptosis bar graph in Figure 2B). It can also be observed that after an initial 8 h phase of increased apoptosis displayed by all control or HIFs-silenced cells, significant differences between control HIFs-expressing and HIFs-silenced cells were revealed at more extended starvation periods (16 or 24 h). Control HIFs-expressing cells seemed to adapt to starvation and decreased apoptosis rate, whereas HIF-silenced, particularly HIF-1α–silenced, cells did not recuperate the basal apoptosis rate after long starvation periods and displayed a statistically significant higher apoptosis rate compared to control HIFs- expressing cells. These results, therefore, suggest that both HIFs, particularly HIF-1α expression, are needed to cope with nutritional stress and to avoid apoptotic cell death, reinforcing the importance of HIF factors in cell survival promotion.

We next investigated if the resistance of colon cancer cells to treatment with cytotoxic agents can be reverted by combining nutritional stress with HIFs knockdown. To this end, we made use of the CCI-779-resistant RKO cell line (Appendix A). The results presented in Figure 3A show how despite incubating cells with a subtoxic dose of CCI-779 (10 μM, which is enough to inhibit mTOR and thus S6 phosphorylation), resistant RKO cells were sensitized to this agent by nutritional stress, either induced by serum starvation or by 8 h incubation in HBSS, since both autophagy (LC3II/I) and apoptosis (cPARP) were increased. However, as can be observed in Figure 3B, combining starvation with HIF-2α knockdown, but not with HIF-1α knockdown, produced a significant enhancement in sensitization to CCI-779 compared to control HIFs-expressing cells, visualized as a substantial increase in the apoptosis rate. It has been reported that HIF-2α silencing produces an elevation of reactive oxygen species (ROS) while the upregulation of HIF-2α lowers intracellular ROS levels [13,23]. In addition, other studies have also demonstrated that HIF-2α counteracts oxidative damage [23,24]. To explore if the observed increase in susceptibility of HIF-2α –silenced cells to CCI-779 and nutritional stress could be mediated by an increase in ROS production, we measured oxidative stress in RKO control and HIFs-silenced cells under normal condition as well as during nutritional stress plus mTORC inhibition. RKO cells were incubated with or without CCI-779 (10 μM) and sensitized to this agent by 8 h starvation in HBSS. Then, cells were incubated 20 min with the oxidation-sensitive fluorescent dye dihydroethidium (DHE) to measure oxidative stress. As it can be observed in Figure 3C, intracellular ROS generation, visualized by an increase in fluorescence intensity resulting from conversion of dihydroethidium to ethidium, was increased as a result of mTORC1 inhibition or by nutritional stress, and was greatly stimulated by the combination of both in control HIFs-expressing cells. In contrast, while this behavior was reproduced in HIFs-silenced cells, the oxidative stress levels were significantly lower in HIF-1α- silenced cells and bigger in HIF-2α-silenced cells than the levels found in control siRNA cells. Because HIF-1α deficient cells display compensatory overexpression of HIF-2α, we investigated if the reduced ROS levels in HIF-1α-silenced cells could be attributed to a HIF-2α -induced protection against oxidative damage. Figure 3D shows how both control siRNA or HIF-1-silenced RKO cells incubated in the presence of the HIF-2α–specific antagonist PT-2385 produced a significant increase in oxidative stress compared with untreated cells, which was blocked by pre-incubation of the cells with the antioxidant compound N-acetylcysteine (NAC), as can be seen in Figure 3E. Taken together, these results suggest that the increased oxidative stress resulting from HIF-2α inhibition/silencing in cells may contribute to sensitize them to nutritional stress and to treatment with cytotoxic agents.

### 2.3. Resistance to Drug Treatment Can also Be Overcome by Only Combining Treatment with HIF-2α Knockdown Expression

It has been reported that chloroquine, an inhibitor of autophagy flux, enhances the effects of 5-FU chemotherapy in human colon cancer cell lines resistant to 5-FU treatment [25,26,27]. We then analyzed the effect of HIFs silencing on cell death induced by this drug in SW480 cells, which are resistant to 5-FU but sensitive to CCI-779 (Appendix A). To this end, control or HIF-silenced cells were treated for 48 h with 5-FU alone, with HCQ alone, or with both 5-FU and HCQ. The results presented in Figure 4 clearly show that 5-FU treatment alone or in combination with HCQ (red arrows in Figure) induced apoptosis resistance in SW480 cells that was not reversed by HIF-1α knockdown. In contrast, HIF-2α knockdown increased apoptotic cell death. It can also be observed that chloroquine alone or in combination increased the LC3II/LC3I ratio, as expected, but did not sensitize cells to death, since apoptosis rates were the same as those in untreated chloroquine cells, either expressing HIFs or not (Figure 4). Taken together, all these data indicate that HIF-2α expression plays a vital role in colon cancer cells in vitro in promoting resistance to stressful environmental conditions such as nutrient deprivation or cytotoxic drug exposure.

### 2.4. HIF-2α Blockade Expression Sensitizes Tumors to Treatment with Autophagy and mTOR Inhibitors In Vivo

To prove in vivo that resistance to drugs can be overcome through coordinated HIF-2α depletion with drug treatment, we made use of a xenograft model in immunocompromised nude mice, as described in Figure 5A. Mice were injected in the left flank subcutaneously (s.c.) with 1 × 10^6^ SW480 cells stably expressing the control siRNA plasmid, and in their right flank, with 1 × 10^6^ SW480 HIF-2α-silenced cells. Although tumors depleted of HIF-2α expression tended to grow more slowly than controls, when they reached approximately 200 mm^3^, mice were randomly segregated into four groups and treatment began. They were injected intraperitoneally with phosphate-buffered saline (PBS, vehicle control), HCQ (65 mg/kg), CCI-779 (0.5 mg/kg) or with the combination of CCI-779 and HCQ. Tumor growth curves were obtained by measuring the tumor size at the indicated time points. The results presented in Figure 5B–D clearly show that HIF-2α knockdown, with or without drug treatment, produced lower tumor volumes in comparison with tumors expressing HIFs in all cases (Figure 5E). Besides, no significant differences were observed between control or drug-treated (HCQ or CCI-779 alone) HIFs-expressing tumors (Control siRNA), but in contrast, there were differences between treatments in mice inoculated with siRNA-HIF-2α-silenced cells. Remarkably, it can also be observed in Figure 5D that the combination of HIF-2α-depleted expression with treatment with both HCQ and CCI-779 inhibitors induced the most significant tumor size decrease in mice (and in several cases, remission as seen in Figure 5E). These results displayed statistical significance compared to HIF-expressing untreated (PBS) tumors, indicating that HIF-2α knockdown sensitized tumor cells most efficiently with the combined treatment of autophagy and mTOR inhibition.

### 2.5. A Specific HIF-2α Heterodimerization Antagonist also Overcame the Cytotoxic Resistance Displayed by Colon Cancer Patient-Derived Primary Culture Cells

Finally, we validated our findings in patient-derived primary colon cancer culture cells displaying resistance to 5-FU or CCI-779 drugs (Appendix A). Transcription factors are typically considered “undruggable”. However, an artificial ligand (PT-2385) that blocks specifically the heterodimerization of HIF-2α with HIF-1β (or ARNT), needed to activate the transcription of genes regulated by HIF-2α [14,28,29,30], has been identified. This ligand acts, therefore, as a HIF-2α-specific dimerization antagonist since it can bind to a cavity only found in the PAS-B domain of HIF-2α and not in the PAS-B domain of HIF-1α, which competes with HIF-2α for dimerizing with HIF-1β at the cell nuclei [14].

To use the inhibitor PT-2385 to validate the results obtained by HIF-2α knockdown, we first examined the concentrations of PT-2385 needed to block the interaction between HIF-2α with HIF-1β. To this end, SW480 cells were incubated for 12 h in the absence or presence of several PT-2385 concentrations. HIF-1β was then immunoprecipitated from cell lysates, and immunoprecipitates were analyzed by Western blotting. Whereas in the absence of PT-2385 both HIF- 1α and HIF-2α co-immunoprecipitated with HIF-1β, in cells treated with 100 μM PT-2385 only HIF-1α coprecipitated with HIF-1β confirming the specificity of the inhibitor [14] and importantly, indicating that concentrations above 50 μM of the inhibitor are required in colon cancer cells. We then examined the effects of several concentrations of 5-FU and CCI-779 on cell viability to confirm the resistance previously found in each patient-derived cell line. As shown in Appendix A, 19739-11K cells and OMCR14-015TK cells displayed resistance against 5-FU treatment, whereas OMCR15-045TK (Appendix A) displayed resistance to both 5-FU and CCI-779. When cells were pre-incubated 12 h in the absence or presence of 100 μM PT-2385 and then in the absence or presence of 5-FU or CCI-779 alone or in combination with HCQ, the inhibition in HIF-2α- mediated transcription resulted in the sensitization of the cells to these drugs, overcoming their resistance to death, as can be observed in Figure 6A–C. It can also be seen that, as previously observed in colon cancer cell lines, and primary cultures obtained from patients, HCQ did not sensitize cells to die alone and in combination with drugs, except in OMCR15-045TK cells, in which the combination of 5-FU with HCQ sensitized cells to death.

Taken together, our data demonstrate that HIFs are essential to promote survival of cancer cells regulating cell death pathways and that drug treatment alone of cancer cells expressing both HIF-1α and HIF-2α quickly induce resistance to apoptotic cell death. In addition, the data shown here indicate that HIF-2α plays a crucial role in promoting drug resistance and cell survival in colon cancer cells.

## 3. Discussion

The development of resistance to treatment of many cancer cells is the main obstacle to overcome in clinical oncology. Oxygen deprivation is a hallmark of solid tumors and the tumor microenvironment. The expression of HIFs is not only crucial for the cellular adaptation to hypoxia but also because HIFs have been associated with resistance promotion and clinical failure [1,2,3,4,31]. The process of autophagy has been recognized as a major regulator of cellular viability under stressful conditions. This process can be triggered in cells by two main mechanisms: (a) HIF-dependent autophagy, induced by HIF-1α/2α, which is mediated by BH3-only proteins BNIP3 and BNIPL3, and promote cell survival in normal or malignant cells [32]; (b) HIF-independent autophagy, positively regulated by AMPK and inhibited by mTORC1, which is induced by metabolic stresses, nutrient depletion and oncogenic activation [33]. Importantly, this type of autophagy in turn can positively modulate the HIF-dependent autophagy, because elevated oncogenic signaling in cancer cells induces HIF-α expression by inducing both their transcription and the stabilization and translation of HIF-α mRNAs [17,32,33,34,35]. Consistent with this, elevated expression of HIF-1α and HIF-2α protein has been observed in a broad array of human cancer cell types in the absence of hypoxia, and we have previously reported and confirmed here that whereas nonmalignant cells do not express HIFs under normoxic conditions, colon cancer cells coexpress HIF-1α and HIF-2α under these conditions [5].

The human RKO and primary SW480 and its derivative metastatic SW620 cell lines used in our study are representative of a range of BRAF-driven and KRAS-driven colorectal genotypes, respectively [15]. We must take into account that in this type of cancer cells HIF-independent autophagy is particularly robust and, therefore, considered as “addicted to autophagy” [17]. In agreement with this, we found in this study that RAS-driven or B-RAF-driven colon cancer cell lines exhibit high basal levels of autophagy compared with nonmalignant cells. However, until now the relevance of the relationship between HIFs and autophagy with the generation of resistance to treatment is unknown.

In this work, we investigated if malignant resistant cells can be sensitized to drug treatment by blocking HIFs expression. To this end, we created a stable knockdown of HIF-1α or HIF-2α in colon cancer cells. Interestingly, we were unable to produce simultaneous knockdown of both HIFs, since we never recuperated clones with double knockdown and all cells finally died (Appendix A). Our results are in agreement with those previously reported by other authors who demonstrated that the loss of one HIF-α subunit is compensated by upregulating the remaining HIF-α isoform, either in cancer cells [12,36,37] or in non-malignant systems such as cartilage [13]. Importantly, other authors have also found that the knockdown of HIF-2α results in a great increase in autophagy levels [12,13], suggesting that lacking one HIF-isoform up-regulates the other and, thereby, elicits survival advantages by dysregulating autophagy. Supporting this, we found here that although it would be expected that the knockdown of each HIF-α isoform, which favors autophagy, would negatively affected autophagy levels, what we obtained was the opposite effect, and that these paradoxical results can be explained by the autophagy induced as a result of the compensatory expression of the remainder HIF-α isoform. However, we can not rule out that the stressful condition imposed by the depletion of one HIF-α isoform could also activate the HIF-independent autophagic pathway and thus the transcription and/or translation of HIFs induced by oncogenic activity, contributing to autophagy induction. In support of this, it is very interesting to note in Appendix A that when we tried to transiently knockdown HIF-1α in stable silenced HIF-2α SW480 cells, surprisingly, the HIF-2α expression was recuperated in only 24 h post-transfection with the sh-HIF-1α plasmid. All these data reinforce the notion that B-RAF and KRAS-driven colon cancer cells are “addicted to autophagy” and indicate, therefore, that HIFs factors play essential roles in regulating cell death pathways to promote cell survival.

It was also found here that colon cancer cells display different sensitivities to drugs and that they become resistant to them despite the combination with the lysosomotropic agent and the autophagy inhibitor hydroxychloroquine (HCQ), which has been reported to synergize with the mTOR inhibitor CCI-779 to suppress melanoma growth and to induce apoptotic cell death [11]. Strikingly, we found that cytotoxic-induced cell death is significantly increased by knockdown of HIFs but not by autophagy inhibition in colon cancer cells. Also, we demonstrated here that although malignant resistant cells can be sensitized to death by nutrient stress, when combined with HIF-2α depletion, they induced severe cell death. Notably, resistance to drug treatment (5-FU or CCI-779, depending on which cell culture was used) was only overcome when combining the treatment with HIF-2α-specific inhibition or knockdown expression, and not with HIF-1α depletion.

Indeed, the main finding obtained here was the demonstration that colon cancer cells displaying different genetic contexts and drug-resistant profiles can be sensitized to treatment by coordinating with HIF-2α depletion expression/inhibition with autophagy inhibition. Strikingly, these results were confirmed in vivo not only in a mouse xenograft model where coordinated autophagy and mTOR inhibition induced tumor remission only in HIF-2α-silenced cells, but importantly, were also confirmed in patient-derived primary colon cancer cells. In this case, patient-derived culture cells corresponding to advanced clinical stages and displaying resistance to treatment, overcame their resistance to 5-FU or CCI-779 when combined with HIF-2α-specific inhibition, thus emphasizing the crucial role played by HIF-2α in promoting resistance and cell survival.

In contrast to the well-established importance of HIF-1α as a robust suppressor of apoptosis, the functional significance of HIF-2α in anti-cancer therapy has been understudied. However, growing experimental evidence has shown that HIF-2α is a critical player in survival promotion. In this regard, we found in this study that HIF-2α, but not HIF-1α, plays a key role in resistance promotion in colon malignant cells. To understand the molecular mechanisms involved, it must be taken into account that although HIF-1α and HIF-2α have some shared gene targets and functions, they also regulate unique genes [5,31]. In this regard, it has been demonstrated that HIF-2α counteracts oxidative damage in cells by inducing the expression of antioxidant enzymes [13,23,24]. In addition, Taniguchi et al. [38] found that overexpressing stabilized HIF-2α, but not HIF-1α, is sufficient to protect mice against radiation-induced gastrointestinal syndrome. In agreement with all this evidence, we showed here that the intracellular oxidative stress levels were significantly increased as a result of HIF-2α -specific inhibition or silencing in colon cancer cells, particularly under nutritional stress and mTORC1 inhibition, suggesting that this may contribute in sensitizing cells to nutritional stress and to treatment with cytotoxic agents, and reinforcing the notion that HIF-2α plays an important cytoprotective role in cancer cells. In addition, it has been demonstrated that HIF-2α, in particular, stimulates autocrine growth signaling and persistent proliferation via the activation of wild-type and mutant key receptor tyrosine kinases such as epidermal growth factor receptor (EGFR) [39,40,41]. Interestingly, Franovic et al. [40] have also reported that genetically diverse cancers develop a common and mandatory program of growth stimulation required for tumorigenesis, at the center of which lies HIF-2α. They demonstrated that the inhibition of HIF-2α blocks the in vivo growth and progression of highly aggressive glioblastoma, non-small-cell lung, and colorectal carcinomas. These authors provided evidence that HIF-2α promotes persistent proliferation by activation of key receptor tyrosine kinases and their major downstream signaling pathways. Consistent with this, Zhou, J. et al. [42] have reported that HIF-2α is primarily responsible for enhancing proliferation, resistance to replication stress and radioresistance in renal cell carcinoma. All this experimental evidence thus suggests that HIF2α inhibition may be effective alone or in combination with therapies other than chemotherapy.

Finally, another explanation of why HIF-1α and HIF-2α play different roles in resistance promotion may rely on the reported existence of a change from HIF-1α to HIF-2α-dependent signaling during cancer progression, which plays a very important role in the promotion of aggressiveness, stemness, and metastasis [31,43,44]. In agreement with this, Koh et al. [43] found that the hypoxia-associated factor (HAF) is an E3 ubiquitin ligase that promotes ubiquitination and degradation of HIF1α. However, interestingly, they showed that HAF binding to HIF2α does not induce its degradation but instead, increases its transactivating activity. Accordingly, the expression of HAF switches the response of the cancer cell to hypoxia from a HIF1α-dependent gene transcription program to another HIF2α-dependent transcription of genes like MMP9 and OCT-3/4, related with the promotion of invasion and cancer stem cell phenotype, associated with highly aggressive tumors in vivo.

In summary, our data demonstrate the reliance upon HIFs and HIF-induced autophagy for survival in aggressive colorectal cancers and indicate that inhibition of HIF-2α, alone or in combination with chemotherapeutic drugs, constitutes a promising strategy to succeed in overcome resistance to cancer treatment.

## 4. Materials and Methods

### 4.1. Reagents and Antibodies

Hydroxychloroquine, temsirolimus (CCI-779), and 5-fluorouracil (5-FU) were purchased from Sigma-Aldrich (St. Louis, MO, USA). The HIF-2α-specific inhibitor PT-2385 was purchased from Biovision. Dihydroethidium (DHE) was obtained from Invitrogen^TM^ (Waltham, MA, USA) (Cat. No. D11347). N-Acetyl-L-cystein (NAC) was obtained from Sigma-Aldrich (Cat. No. A7250). The antibodies used in the experiments were from the following sources: anti-LC3 from Novus Biologicals (Centennial, CO, USA); and anti-activated caspase-3, anti-HIF-1α, anti-HIF-2α, anti-SQSTM1/p62, anti-β-tubulin, anti-Atg7, anti-Phospho-S6 Ribosomal protein (Ser235/236), anti-Beclin.1 and anti-PI3K-Class III were all obtained from Cell Signaling Technology (Danvers, MA, USA). Goat anti-mouse and anti-rabbit IgG-horseradish peroxidase-conjugates were from Pierce (Rockford, IL, USA)

### 4.2. Ethics Statement

This work has been conducted following the ethical standards according to the Declaration of Helsinki and according to national and international guidelines and has been approved by the Faculty of Medicine Ethical Committee at the Universidad Nacional Autónoma de México (in accordance to the Mexican Official Norm NOM-062-ZOO-1999).

### 4.3. Plasmids

The control plasmid containing a scrambled shRNA sequence was obtained from Santa Cruz Biotechnology. The control plasmid (void pSuper), HIF-1α, and HIF-2α RNAi plasmids were donated by Dr. Daniel Chung, and their construction and effectiveness were described previously [45]. The reporter plasmid encoding EGFP-LC3 was obtained from Addgene (Watertown, MA, USA) (plasmid #11546), a non-profit organization dedicated to facilitating plasmid sharing among scientists.

### 4.4. Cell Lines

The following colorectal cancer cell lines were used. The human RKO malignant cells display normal canonical Wnt signaling (expressing wild-type APC protein) and are the prototype of BRAF-driven cancer cells (B-raf V600E and PIK3CA H1047R mutations) [15]. The human carcinoma SW480 cell line expresses a truncated version of APC, has constitutively active canonical Wnt signaling, and is the prototype of KRAS-driven cancer cells (KRAS G12V, APC A1457T/K1462R, FGFR3 S400R, TP53 R273H, and STK11 G58S mutations) [15]. All cancer cell lines and the non-malignant 112CoN cell line used here were obtained from American Type Culture Collection (ATCC; Manassas, VA, USA) and were authenticated in June 2017 by Short Tandem Repeat DNA profiling were done at the Instituto Nacional de Medicina Genómica (INMEGEN) in Mexico City.

All cells were cultured in a humidified 5% CO_2_ incubator at 37 °C. For starvation cells were washed twice with phosphate buffer saline (PBS, GIBCO/Invitrogen; Waltham, MA, USA) and placed in HBSS buffer (GIBCO/Invitrogen).

### 4.5. Primary Cell Cultures Derived from Colorectal Cancer (CRC) Patients’ Tissue Specimens

Tissue samples were obtained from CRC patients undergoing therapeutic intervention at the Unit of Oncological Surgery, IRCCS-AOU San Martino-IST, Genoa, provided informed consent (the study was approved by the institutional and regional ethical committee, PR163REG2014). CRC specimens were minced with scissors, transferred into 15-ml conical tubes and digested with 2 mg/ml collagenase type I and II (Sigma-Aldrich, Darmstadt, Germany) in RPMI 1640 (Gibco, Monza, Italy) for 45 min at 37 °C. Residual tissue debris was removed by soft centrifugation (300 rpm, 1 min), cells were pelleted (1800 rpm, 10 min) and passed through a 100 μm cell strainer (Euroclone, Milan, Italy).

The OMCR15-045TK epithelial cell line was derived from a stage IIA (UICC 2009) CRC, showing a strong peri and intratumoral infiltration of lymphocytes. The 19739-11 K epithelial cell line was derived from stage IIIA showing the KRAS G12D mutation and the OMCR14-015TK epithelial cell line was derived from a stage IV with multiple metastases. After collagenase digestion, intact crypts collected from the pellet of residual tissue debris were plated in DMEM/F12 with Hepes buffer (Euroclone) containing B27 supplement, EGF 10 ng/mL and DTT 10 nM (Sigma).

### 4.6. Western Blotting and Apoptosis Analysis

Protein samples (30 μg) were separated by 10% or 15% sodium dodecyl sulfate polyacrylamide gel electrophoresis (SDS-PAGE) followed by electrophoretic transfer onto nitrocellulose membranes (Bio-Rad, Hercules, CA, USA). The membranes were blocked with 5% nonfat dry milk and incubated overnight at 4 °C with the corresponding primary antibody. Detection was performed using the SuperSignal Kit (Pierce) with a horseradish peroxidase-conjugated second antibody. An actin antibody or a β-tubulin antibody was used as the control for equal loading.

Apoptotic cell death was examined by Western blotting to detect the presence of either cleaved caspase-3 or cleaved PARP proteins.

### 4.7. Autophagy Detection

Detection of autophagy in live cells was performed by flow cytometry using the CYTO-ID autophagy detection kit (Enzo Life Sciences, Farmingdale, NY, USA) according to the manufacturer’s instructions, in the absence or presence of hydroxychloroquine (HCQ) to inhibit autophagy flux.

### 4.8. HIF-1α or HIF-2α Knockdown

To induce stable silencing of HIF-1α or HIF-2α, the cells were transfected with the pSuper HIF-1α or HIF-2α RNAi plasmids, which were constructed and analyzed by Dr. Daniel Chung as previously described [5,45] or with the control plasmid (encoding a scrambled shRNA sequence or pSuper void plasmid) using Lipofectamine 2000 (Invitrogen^TM^, Waltham, MA, USA). Cells were transfected with either 1 μg of the control plasmid or 1 μg of pSuper HIF-1α RNAi or HIF-2α RNAi plasmids. Stable HIF-1α RNAi or HIF-2α RNAi transfectants were selected with 3 μg/mL puromycin (Sigma) or 5 μg/mL G418 (Sigma), respectively, during four weeks, and the clones were selected and screened for HIF-1α or HIF-2α silencing by flow cytometry and Western blotting.

### 4.9. Flow cytometry/ Fluorescence Activated Cell Sorting (FACS) Analysis

Briefly, cells were detached and dissociated in 10 mM EDTA solution. Then cell suspension was washed, and resuspended in PBS supplemented with 4% fetal calf serum (staining buffer). The cells were incubated with the corresponding primary and secondary antibodies. The cells stained with the secondary antibody alone were used as a negative control. Cells were acquired in an Attune Nxt (Thermo Scientific, Waltham, MA, USA) and data were analyzed with the software FlowJo (Tree Star^®^, Ashland, OR, USA).

### 4.10. Viability Assay

The cell viability was measured with the 3-(4,5-dimethylthiazole-2-yl)-2,5-biphenyl tetrazolium bromide (MTT) assay. SW480 cells were grown in the absence or presence of different doses of CC1-779, HCQ or 5-FU, alone and in combination for 24, 48 or 72 h. Then the cells were incubated with MTT (0.5 mg/mL) during the last 3 h of each incubation period at 37 °C. The reaction was stopped adding 0.1 mL of acid isopropanol to each well. Formazan salts were dissolved and quantified by spectrophotometry at 570 nm.

### 4.11. Immunofluorescence Analysis

Control or HIF-1α- or HIF-2α knockdown cells were transiently transfected with 2 μg EGFP-LC3 plasmid (ptfLC3, Addgene) using Lipofectamine 2000 following the manufacturer’s instructions. The cells were grown on glass-bottom Petri dishes, and LC3 fluorescence was analyzed using a confocal microscope (Leica TCS SP5, Morrisville, NC, USA) with a krypton-argon laser. The images were analyzed for quantification with the Image J program version 1.47b obtained from the National Institutes of Health website (http://imagej.nih.gov/ij/).

### 4.12. Xenograft Tumor M Model

The HCQ and CCI-779 doses selected for these experiments in vivo were chosen according to the dose reported for mice in xenograft experiments [11] and on the human dose of 200 mg HCQ twice daily, commonly used in clinical practice that has been escalated [11,46]. Six-week-old male nude mice were s.c. injected in their left flank with 1 × 10^6^ SW480 cells stably expressing the control siRNA plasmid, and in the same animals, in their right flank, with 1 × 10^6^ SW480 HIF-2α-silenced cells. Once tumors reached approximately 200 mm^3^, mice were randomly segregated into four groups (5 mice per group and two xenograft tumors per mouse) and treated by an intraperitoneal injection of PBS (vehicle control), HCQ (65 mg/kg), CCI-779 (0.5 mg/kg) or the combination of CCI-779 and HCQ, respectively. Tumor growth curves were determined by measuring the tumor size at the indicated time points.

### 4.13. Analysis of Oxidative Stress

To measure intracellular ROS production, the culture media was carefully removed and replaced by fresh medium supplemented with 5 nM of dihydroethidium (DHE). The cells were incubated for 20 min at 37 °C protected from light. The medium was removed, the cells were washed and fixed with 1% of paraformaldehyde. The cells were co-stained with 4′6-diamidino-2-phenylindol (DAPI) for nuclear staining. The images were captured using an IX71 Inverted confocal fluorescent Microscope (Olympus, Tokyo, Japan) with excitation/emission of 567/610 nm. The fluorescence intensity of 50 cells obtained from at least 8 different view fields for each condition was quantified using the Image-J program. To evaluate the oxidative stress in the absence or presence of an antioxidant agent, N-Acetyl-L-cystein (NAC) was used. Briefly, the culture media was removed and fresh medium supplemented with 3 mM of NAC was added. The cells were incubated during 2 h at 37 °C before incubation of each condition as indicated.

### 4.14. Statistical Analysis

The data are expressed as the mean ± standard error of the mean (SEM). Statistical data analysis was performed using one-way analysis of variance (ANOVA) with Tukey’s or Bonferroni’s multiple comparison tests and was performed with the GraphPad Prism program. A *p*-value < 0.05 was considered statistically significant.

## 5. Conclusions

Our results indicate that the most efficient way to sensitize human colon cancer-resistant cells to treatment is through the specific inhibition of HIF-2α or by blocking its expression. In addition, our data clearly show that although both HIF-1α and HIF-2α are involved in survival promotion, they participate in different ways to regulate cell death pathways in colon cancer cells and point to blockade of HIF-2α-mediated actions alone or in combination with drugs as a compelling approach to improve therapeutic outcomes in advanced CRC.

## Figures and Tables

**Figure 1 cancers-11-00755-f001:**
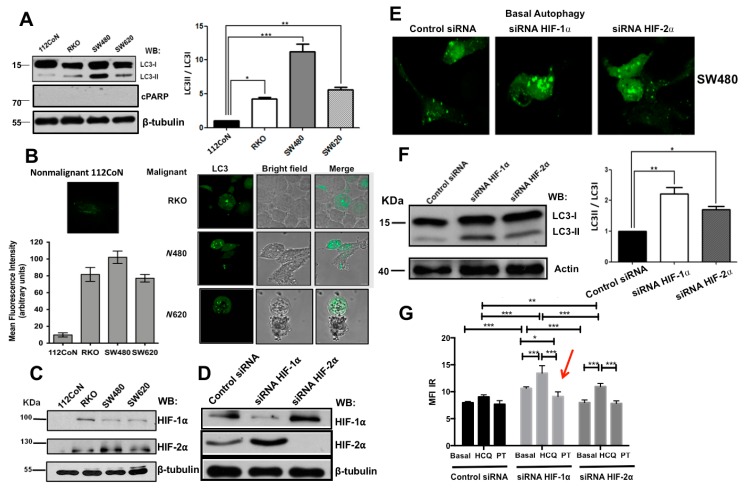
(**A**) RAS-driven and BRAF-driven human colon cell lines show higher levels of basal autophagy than non-malignant cells. The expression of the ratio LC3II/I was examined in colon non-malignant 112CoN or colon malignant RKO, SW480, and SW620 cells by Western blotting. The β-tubulin antibody was used to control for equal loading. Densitometric analysis was performed to estimate the changes in LC3II7LC3I ratio levels in cancer cells compared with 112CoN non-malignant colon cells; the bar graphs represent the means ± standard error of the mean (SEM) from at least three independent assays. * *p* < 0.05; ** *p* < 0.01; *** *p* < 0.001. (**B**) Colon non-malignant 112CoN or colon malignant RKO, SW480, and SW620 cells were transiently transfected with an EGFP-LC3 -expressing plasmid and grown in normal medium. Twenty-four hours after transfection, the cells were visualized by the presence of LC3 puncta under the confocal microscope (Leica TCS SP5) with a krypton-argon laser. The images (40×) were analyzed for quantification with the Image J program. Data shown are representative of three independent experiments. (**C**) Analysis of hypoxia-inducible factors (HIFs) expression in colon cancer cells compared with non-malignant cells by immunoblotting. Total cell extracts from colon cell lines were prepared, and the samples were subjected to sodium dodecyl sulfate polyacrylamide gel electrophoresis (SDS-PAGE) followed by immunoblotting using the indicated antibodies. β-tubulin was used as a control for equal loading. Data are representative of three independent experiments. (**D**) Knockdown efficiency of HIF expression analysis was performed as described in “Materials and Methods”. Results shown are representative of three independent experiments using different cell preparations. (**E**–**G**) Basal autophagy levels are increased as a result of HIFs depletion expression. Stable control or HIFs-silenced SW480 cells were transiently transfected with an EGFP-LC3 expressing plasmid and grown in glass-bottom Petri dishes in normal medium. 24 h after transfection, autophagosomes were visualized for the presence of LC3 puncta by laser confocal microscopy (40×). (**F**) The expression of the ratio LC3II/LC3I was examined in stable control or HIFs-silenced SW480 cells by Western blotting. Actin antibody was used to control for equal loading. Densitometric analysis was performed to estimate the changes in LC3II/LC3I ratio levels in HIF-silenced SW480 cells compared with control SW480 HIF-expressing cells.; the bar graphs represent the means ± SEM from at least three independent assays. * *p* < 0.05; ** *p* < 0.01. (**G**) Detection of autophagy in stable live control or HIFs-silenced SW480 cells incubated in the absence or presence of 100 μM PT-2385 (HIF-2α antagonist) was performed by flow cytometry using the CYTO-ID Autophagy detection kit, in the absence (basal) or presence of the autophagy flux inhibitor hydroxychloroquine (HCQ). The bar graph represents the mean ± SEM of three independent experiments. * *p* < 0.05; ** *p* < 0.01; *** *p* < 0.001.

**Figure 2 cancers-11-00755-f002:**
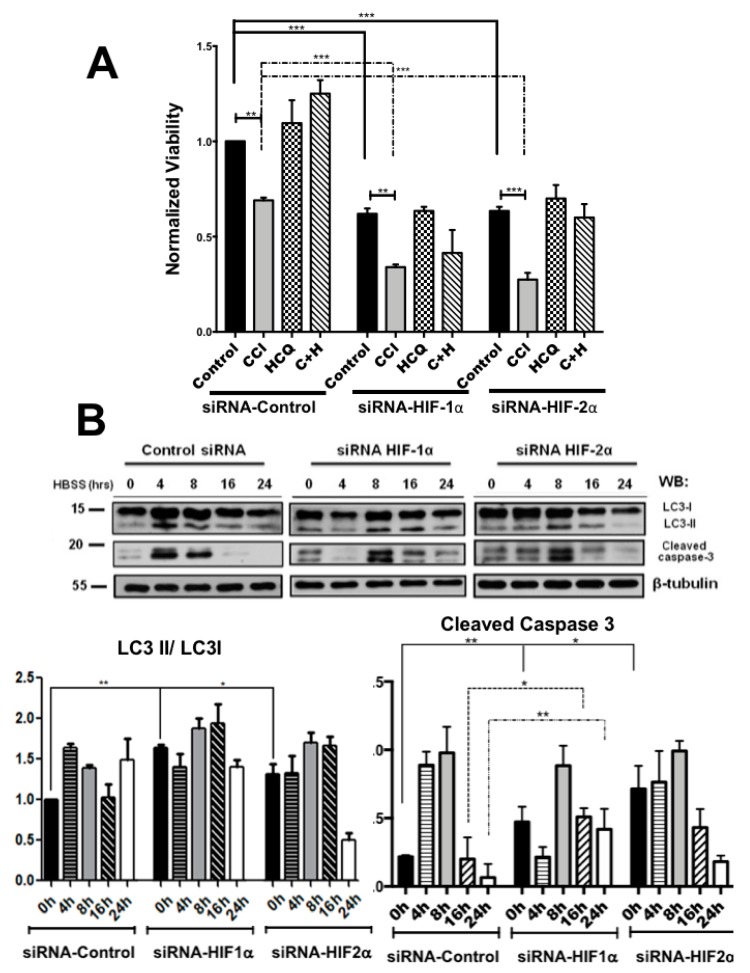
(**A**) Cell viability diminishes mainly as a result of HIFs-silencing expression. SW480 cells were incubated 48 h in the absence or presence of several concentrations of the drugs (the mTORC1 inhibitor CCI-779 or the autophagy inhibitor hydroxychloroquine) alone or in combination, as indicated in the figure. The cell viability was measured with the MTT assay as described in the Materials and Methods section. The bar graph represents the mean ± SEM of three independent experiments. ** *p* < 0.01; *** *p* < 0.001. (**B**) Nutritional stress-induced autophagy in both HIF-expressing or HIF-silenced SW480 cells, but both autophagy and apoptosis rates were bigger in HIFs- depleted cells than in controls. Growth medium from control or HIF-silenced cells was replaced with Hanks’ balanced salt solution (HBSS). Cells were collected at the time points indicated in the Figure and examined by Western blotting to detect autophagy and apoptosis as indicated in the Figure. β-tubulin was used as a control for equal loading. The results shown are representative of at least three independent experiments. Densitometric analysis was performed to estimate the changes in LC3II/I ratio and in cleaved Caspase 3 levels compared to the levels found in control HIFs expressing cells (bar graphs) and data represents the means ± SEM of at least three independent experiments. * *p* < 0.05; ** *p* < 0.01.

**Figure 3 cancers-11-00755-f003:**
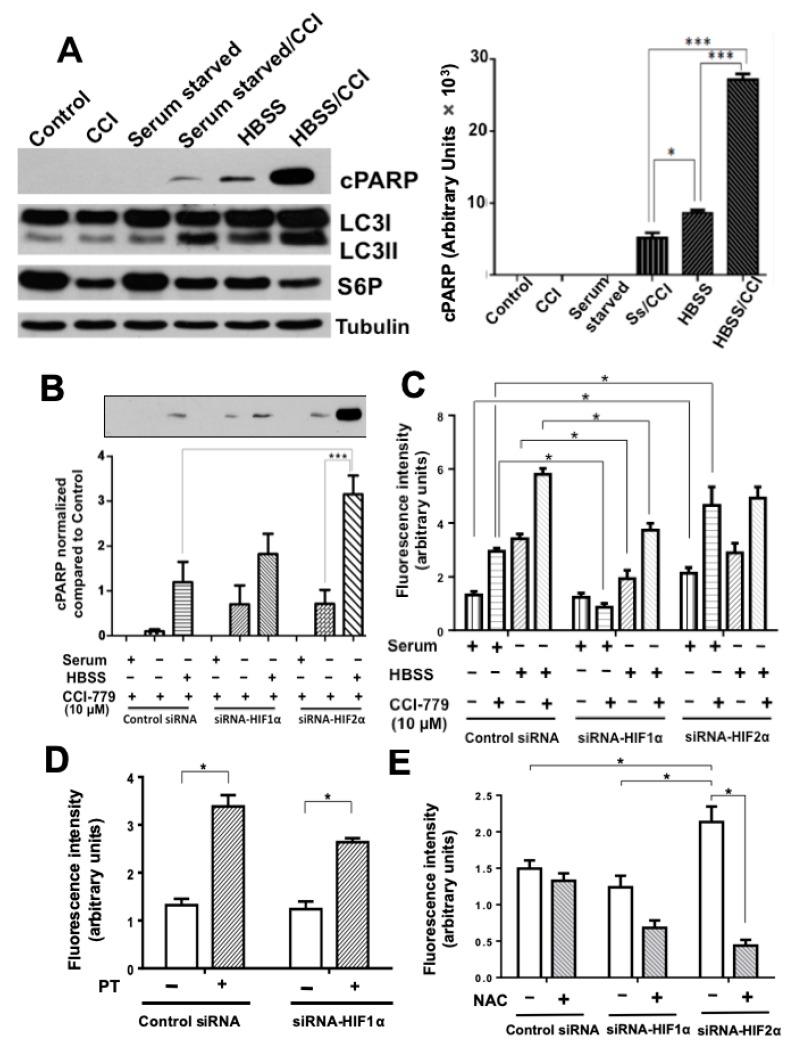
CCI-779-resistant RKO cells are also greatly sensitized to this mTORC1 inhibitor by combining nutritional deprivation with HIF-2α depletion. (**A**) Cells were grown in the absence or presence of serum (8 h), in the presence of HBSS during 8 h, in the presence of 10 μM CCI-779, or in combination during 8 h. Cell extracts were analyzed by Western blotting to detect the expression of LC3II/LCI ratio, cleaved PARP, and phosphorylated ribosomal protein S6 (as control of mTOR efficiency inhibition). β-tubulin was used as a control for equal loading. The results shown are representative of three independent experiments. The densitometric analysis shows the changes in LC3II/I ratio and in cleaved PARP levels compared to the levels found in basal (untreated) control HIFs-expressing cells. Bar graphs represent the means ± SEM of at least three independent experiments. * *p* < 0.05; *** *p* < 0.001. (**B**) Control or HIFs-silenced cells were grown in the absence or presence of serum, HBSS or 10 μM CCI-779 alone, or in combination, during 8 h. The presence of apoptosis by cleaved PARP detection was examined in each condition by Western blotting (panel B). Densitometric analysis was performed to estimate the changes in cleaved PARP levels compared to the levels found in controls. The bar graph shows the means ± SEM from at least three independent experiments. *** *p* < 0.001. (**C**–**E**) HIF-2α depletion increases oxidative stress. (**C**) Control or HIFs-silenced cells were grown in the absence or presence of serum, HBSS or 10 μM CCI-779 alone, or in combination, during 8 h. Oxidative stress was measured by incubating the cells with 5 nM of the oxidation-sensitive fluorescent dye DHE for 20 min at at 37 °C in darkness. Cells were washed, fixed and the fluorescence intensity measured using an inverted confocal fluorescent microscope. The bar graph shows the means ± SEM of fluorescence intensity of 50 cells obtained from at least 8 different view fields for each condition. * *p* < 0.01. (**D**) Control or HIF-1α-silenced cells were grown during 16 h in the absence or presence of 100 μM of the HIF-2α -specific antagonist PT-2385, and oxidative stress was then measured as described. (**E**) Control or HIFs-silenced cells were incubated in the absence or presence of 3 mM of the antioxidant *N*-acetylcysteine (NAC) during 2 h. Then oxidative stress was measured as described.

**Figure 4 cancers-11-00755-f004:**
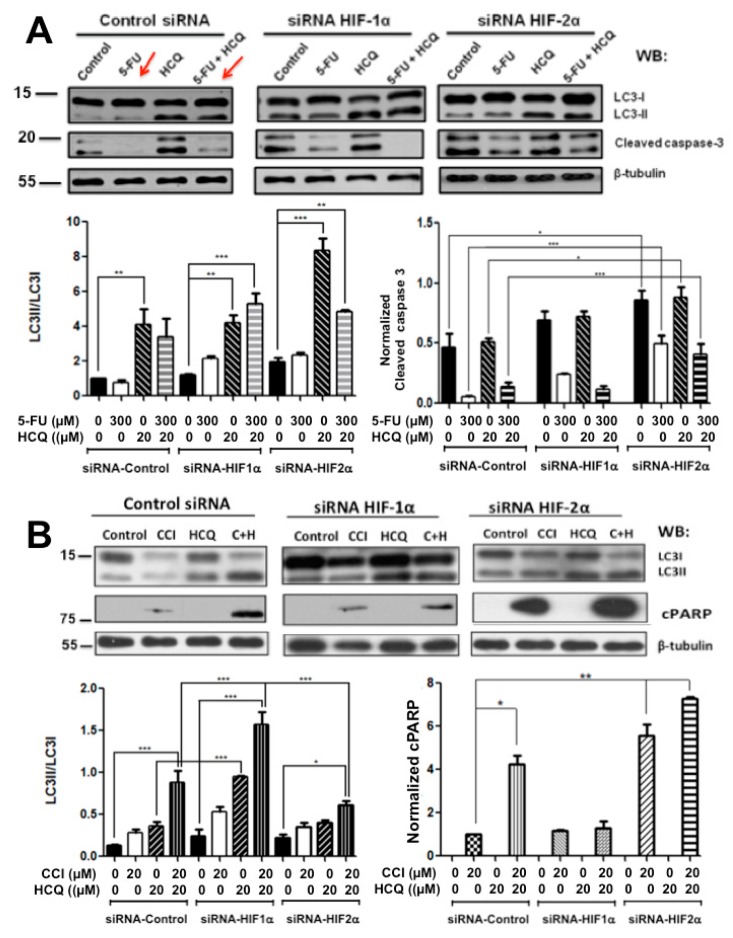
Comparative analysis of the effect of HIFs silencing on cell death induced by drug treatment produced in sensitive vs. resistant SW480 colon cancer cells. (**A**) Resistance against 5-Fluoruracil (5-FU) can be reversed by HIF-2α silencing. Control or HIF-silenced SW480 cells, which are resistant to 5-FU, were treated with 300 μM 5-FU alone, with 20 μM HCQ, or in combination with both agents for 48 h. Cell extracts were analyzed by Western blotting to detect autophagy levels and apoptosis. β-tubulin was used to control for equal loading. (**B**) Control or HIF-silenced SW480 cells, which are sensitive to the mTORC1 inhibitor CCI-779, were treated with 20 μM CCI-779 alone, with 20 μM HCQ, or in combination with both agents for 48 h. Cell extracts were analyzed by Western blotting to detect autophagy levels and apoptosis. β-tubulin was used as a control for equal loading. The results shown in both panels are representative of at least three independent experiments using different cell preparations. Densitometric analysis was performed in each panel to estimate the changes in LC3II/I ratio and in cleaved Caspase 3 levels compared to the levels found in control HIFs- expressing cells. Data represent the means ± SEM of at least three independent experiments. * *p* < 0.05; ** *p* < 0.01; *** *p* < 0.001.

**Figure 5 cancers-11-00755-f005:**
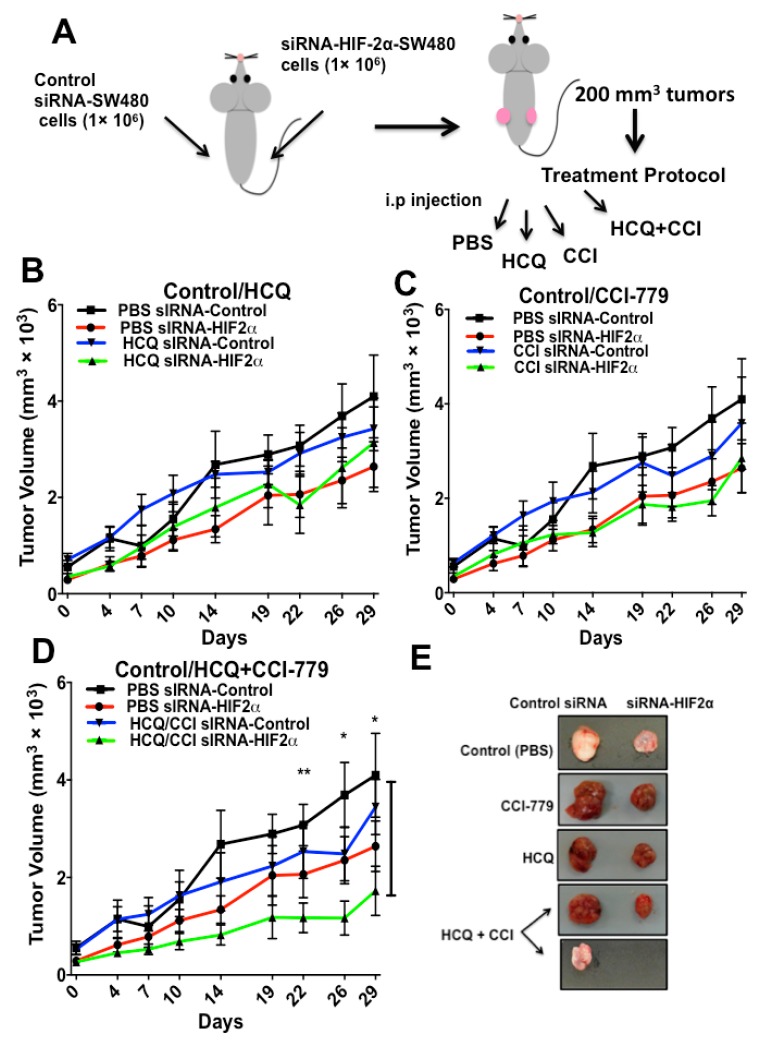
Resistance to drugs can be overcome through coordinate HIF-2α depletion with autophagy inhibition and drug treatment in vivo using a mouse xenograft model. (**A**) Six-week-old male nude mice were s.c. injected at their left flanks with 1 × 10^6^ SW480 cells stably expressing the control siRNA plasmid, and in the same animals, at their right flanks, with 1 × 10^6^ SW480 HIF-2α -silenced cells. Once tumors reached around 200 mm^3^, mice were randomly segregated into four groups (5 mice per group and two xenograft tumors per mouse) and began treatment: they received i.p injection of phosphate-buffered saline (PBS, control), or HCQ (65 mg/kg), or CCI-779 (0.5 mg/kg) or the combination of CCI-779 and HCQ, respectively. (**B**–**D**) Tumor growth curves were obtained by measuring the tumor volumes at the indicated time points. Each data point represents the mean ± SEM tumor volume from 5 mice in each condition. * *p* = 0.05; ** *p* = 0.01 (*t*-test) compared with vehicle (PBS) control. (**E**) Photographs of the representative tumors obtained in each experimental condition tested.

**Figure 6 cancers-11-00755-f006:**
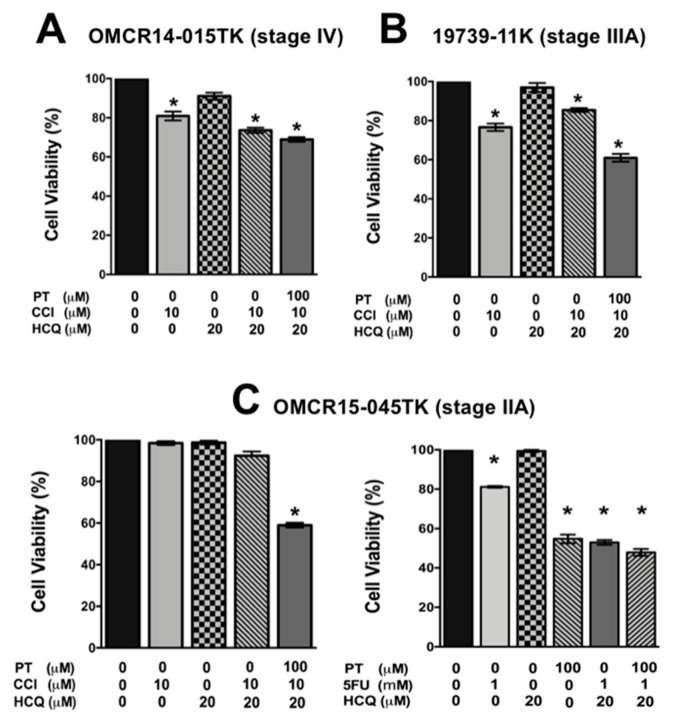
A specific HIF-2α heterodimerization antagonist (PT-2385) also overcame the cytotoxic resistance displayed by colon cancer patient-derived primary culture cells. (**A**–**C**) OMCR14-015TK (**A**), 19739-11K (**B**) or OMCR15-045TK (**C**) patient-derived cells representing different clinical stages were incubated 48 h in the absence or in the presence of several concentrations of the drugs, alone or in combination, as indicated in the Figure. The cell viability was measured by the MTT assay as described in the Materials and Methods section. The bar graph represents the mean ± SEM of three independent experiments. * *p* < 0.05.

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
