# Peer review of "Functional Interaction of Hypoxia-Inducible Factor 2-Alpha and Autophagy Mediates Drug Resistance in Colon Cancer Cells"

_cancers, 2019, doi:10.3390/cancers11060755_

Round 1

Reviewer 1 Report

As mentioned in my previous comments, data presented in this study are interesting. Importantly, this study proposes the use of HIF2 inhibitors in combination with other pharmacological interventions to counteract tumor growth. However authors have not provided experimental evidence related the HIF2a-dependent molecular mechanisms underlying their findings. Their novel experimental approaches (not shown in the response to reviewers letter) have ruled out the involvement of mTORC1 activity, which was suggested in my previous comments. Moreover authors propose that the increased autophagy is not involved in the reduced viability of HIF1 and HIF2-silenced cells (shown in Figure 2A). Therefore what are the HIF2-dependent mechanisms involved? In my previous comments it was suggested the following “Authors should measure intracellular oxidative stress in control and HIF2-silenced cells under normal conditions as well as during nutritional stress plus CCI-779 treatment. In this line, authors should assess whether HIF2a-silenced cells show a reduced expression of antioxidant enzymes analyzed in the study referred above (PMID: 19706526). If HIF2-silenced cells show higher oxidative stress, authors should attempt to recover their viability using antioxidant treatment”. However authors have not addressed the involvement of oxidative stress as a potential explanation of their findings. Authors might assess this suggested oxidative mechanisms or any other potential mechanisms that can provide molecular basis of their data.     

In addition, authors show that induction autophagy in HIF1a-silenced is mediated by the compensatory activation of HIF2a isoform (Figure 1G). These data suggest that HIF2a favors autophagy in these cells. However authors show that silencing of HIF2 elevates the LC3-II / LC3-I ratio (Figure 1F) but not using flow cytometry (Figure 1G). Authors should clarify whether silencing of HIF2a induces autophagy or not and if this is the case how can be understood that inhibition of HIF2a with PT-2385 and silencing of HIF2a show opposite effects on cellular autophagy.

Finally the following study PMID: 19273585 should be included in the reference section.

Author Response

1. As mentioned in my previous comments, data presented in this study are interesting. Importantly, this study proposes the use of HIF2 inhibitors in combination with other pharmacological interventions to counteract tumor growth.  However authors have not provided experimental evidence related the HIF2a-dependent molecular mechanisms underlying their findings. Their novel experimental approaches (not shown in the response to reviewers letter) have ruled out the involvememnt of mTORC1 activity, which was suggested in my previpus comments.  Moreover authors propose that the increased autophagy is not involved in the reduced viability of HIF1 and HIF2-silenced cells (shown in Figure 2A). Therefore what are the HIF2-dependent mechanisms involved?  In my previous comments it was suggested the following; “Authors should measure intracellular oxidative stress in control and HIF2-silenced cells under normal conditions as well as during nutritional stress plus CCI-779 treatment. In this line, authors should assess whether HIF2a-silenced cells show a reduced expression of antioxidant enzymes analyzed in the study referred above (PMID: 19706526). If HIF-2-silenced cells show higher oxidative stress, authors should attempt to recover their viability using antioxidant treatment”.  However, authors have not addressed the involvement of oxidative stress as a potential explanation of their findings. Authors might assess this suggested oxidative mechanisms or any other potential mechanisms that can provide molecular basis of their data.

Fortunately, we had followed your suggestion since your previous revision, because it imply much work that cannot be completed within 10 days, the time gave to us now in the Editorial report to re-submit our work. As you requested, these results are now included in the manuscript. We have explored if the observed increase in susceptibility of HIF-2α –silenced cells to CCI-779 and nutritional stress could be mediated by an increase in ROS production, measuring oxidative stress in RKO control and HIFs-silenced cells under normal condition as well as during nutritional stress plus mTORC inhibition.  As we have explained in the Results section, cancer cells were incubated with or without mTORC1 inhibitor (CCI-779) and sensitized to this agent by 8 h starvation in HBSS. Then, cells were incubated with the oxidation-sensitive fluorescent dye dihydroethidium (DHE) to measure oxidative stress. As it can be observed in the new Figure 3C, intracellular ROS generation was increased as a result of mTORC1 inhibition or by nutritional stress, and was greatly stimulated by the combination of both in control HIFs -expressing cells. In contrast, while this behavior was reproduced in HIFs-silenced cells,  the oxidative stress levels were  significantly  lower in HIF-1a- silenced cells and bigger in HIF-2a -silenced cells than the levels found in control siRNA cells.  Because HIF-1a deficient cells display compensatory overexpression of HIF-2a, we investigated if the reduced ROS levels in HIF-1a silenced cells could be attributed to a HIF-2a -induced protection against oxidative damage. New Figure 3D shows how both control siRNA or HIF-1a-silenced RKO cells incubated in the presence of the HIF-2a–specific antagonist PT-2385 produced a significant increase in oxidative stress compared with untreated cells, which was blocked by pre-incubation of the cells with the antioxidant compound N-acetylcysteine (NAC), as it can be seen in new Figure 3E.  Taken together, these results suggest that the increased oxidative stress resulting from HIF-2a inhibition/silencing in colon cancer cells contribute to sensitize them to nutritional stress and to treatment with cytotoxic agents.  

 Thus, regarding your question “what are the HIF2-dependent mechanisms involved?”  We have explained  in the Discussion section the following:

“We found in this study that HIF-2α, but not HIF-1α, plays a key role in resistance promotion in colon malignant cells. To understand the molecular mechanisms involved, it must be taken into account that although HIF-1α and HIF-2α have some shared gene targets and functions, they also regulate unique genes [5, 31]. In this regard, it has been demonstrated that HIF-2α counteracts oxidative damage in cells by inducing the expression of antioxidant enzymes (13. 23. 24). In addition, Taniguchi et al. [38] found that overexpressing stabilized HIF-2α, but not HIF-1α, is sufficient to protect mice against radiation-induced gastrointestinal syndrome. In agreement with all this evidence, we showed here that the intracellular oxidative stress levels were significantly increased as a result of HIF-2a -specific inhibition or silencing in colon cancer cells, particularly under nutritional stress and mTORC1 inhibition, suggesting that this importantly contribute in sensitizing cells to nutritional stress and to treatment with cytotoxic agents, and reinforcing the notion that HIF-2α plays an important cytoprotective role in cancer cells.  In addition, it has been demonstrated that HIF-2α, in particular, stimulates autocrine growth signaling and persistent proliferation via the activation of wild type and mutant key receptor tyrosine kinases such as epidermal growth factor receptor (EGFR) [39-41]. Interestingly, Franovic et al. [40] have also reported that genetically diverse cancers develop a common and mandatory program of growth stimulation required for tumorigenesis, at the center of which lies HIF-2α. They demonstrated that the inhibition of HIF-2α blocks the in vivo growth and progression of highly aggressive glioblastoma, nonsmall-cell lung, and colorectal carcinomas. These authors provided evidence that HIF-2α promotes persistent proliferation by activation of key receptor tyrosine kinases and their major downstream signaling pathways.  Consistent with this, Zhou J et al [42] have reported that HIF-2α is primarily responsible for enhancing proliferation, resistance to replication stress and radioresistance in renal cell carcinoma. All this experimental evidence thus suggests that HIF-2α inhibition may be effective alone or in combination with therapies other than chemotherapy.

Finally, another explanation of why HIF-1α and HIF-2α play different roles in resistance promotion may rely on the reported existence of a change from HIF-1α to HIF-2α-dependent signaling during cancer progression, which plays a very important role in the promotion of aggressiveness, stemness, and metastasis [31, 43, 44]. In agreement with this, Koh et al. [43] found that the hypoxia-associated factor (HAF) is an E3 ubiquitin ligase that promotes ubiquitination and degradation of HIF-1α. However, interestingly, they showed that HAF binding to HIF-2α does not induce its degradation but instead, increases its transactivating activity. Accordingly, the expression of HAF switches the response of the cancer cell to hypoxia from a HIF1α – dependent gene transcription program to another HIF2α –dependent transcription of genes like MMP9 and OCT-3/4, related with the promotion of invasion and cancer stem cell phenotype, associated with highly aggressive tumors in vivo”.

2. Authors show that induction autophagy in HIF1a-silenced is mediated by the compensatory activation of HIF2a isoform (Figure 1G).  These data suggest that HIF2a favors autophagy in these cells. However authors show that silencing of HIF2 elevates de LC3-II/ LC3-I  ratio (Figure 1F) but not using flow cytometry (Figure 1G). Authors should clarify whether silencing of HIF2a induces autophagy or not and if this is the case how can be understood  that inhibition of HIF2a with PT-2385 and silencing of HIF2a show opposite effects on cellular autophagy.

Your are right pointing out that the results showed in Figure 1E (immunofluorescence analysis) and Figure 1F (Western blotting analysis) clearly  show that silencing of HIF-2α increased autophagy levels compared with control siRNA cells, but that we did not obtain exactly the same using flow cytometry (Figure 1G). However, your observation apply for the autophagy basal levels and PT-2385 levels found in HIF-2α-silenced cells,  which by flow cytometry did not show significant differences compard to the levels found in siRNA control cells. However, when we used the autophagy inducer hydroxychloquine, the autophagy levels were significantly increased compared to control cells in both HIF-1α and HIF-2α- silenced cells (Figure 1G). Most importantly, the fact that the specific HIF-2α antagonist PT-2385 reverted (indicated with a red arrow in Figure 1G) the increase in autophagy levels produced as a result of HIF-1α –silencing, which displayed a compensatory HIF-2α overexpression, indicated that the increased autophagy was mediated by HIF-2α.  Therefore, taken together our data presented in Figure 1E, F and G, clearly show that HIF-2α silencing induces an increase in autophagy levels.  But how to explain why Western blotting and flow cytometry did not display the same differences in autophagy basal levels between control and HIF-2α knockdown cells, or between untreated or PT-2385 treated cells using flow cytometry?   In first place, it must be taken into account that although we can measure autophagy by Western blotting or by flow cytometry, these techniques possess different sensibilites and disadvantages. By Western blotting it is posibble to observe and quantify the appearance of the autophagy-specific  marker LC3 II, whereas by flow cytometry it is not and we can only measure the amount of autophagosomes present in the cells. Besides, in flow cytometry there is also the problem related to the presence of  high levels of cell autofluorescence, which in this case, is complicated not only by the elevated basal autophagy characteristically found in cancer cells, but also by the use of a fluorescent compound to measure autophagosomes. We think that by these reasons, we did not observe significant differences between untreated or PT-2385 treated HIFs expressing or HIFs-silenced cells, but in any case we did not observe opposite effects induced by HIF-2α silencing or by HIF-2α inhibition -mediated by PT-2385.

3. Finally the following study PMID: 19273585 should be included in the reference section.

This study has been now included in the Reference section  (reference number 32).

Reviewer 2 Report

The authors found increased autophagy and increased HIF-1a and HIF-2a in colorectal cancer cells as compared to non-malignant cells. Knocking down HIF-1a OR HIF-2a increases autophagy.

The authors then found that cell viability decreases when either HIF-a is knocked down. Theree was not a synergistic effect (just additive) with mTOR inhibition, autophagy inhibition or a cytotoxic drug.

However, when HIF-2a knockdown is combined with starvation cells were sensitized to a cytotoxic drug. This was not seen with HIF-1a knockdown.

Combining autophagy inhibition, with HIF-2a knockdown and cytotoxicity increases apoptotic death (HIF-1a knockdown does not induce this), indicating HIF-2a is contributing to drug resistance.

Inhibition of HIF-2a in xenograft mice increases the sensitivity of tumours to autophagy and mTOR inhibitors.

Overall an interesting paper that needs some minor corrections (notes) with two things that must be corrected before publication.

1) Supplementary Figure 5- This figure and the interpretation are not acceptable in current form. In the left blot, furthest left lane there is a very dark band at the bottom. My interpretation would be that this is the HIF-1a band but the authors claim that a higher (and very, very faint band) is actually the HIF-1a band. Either the antibody is VERY non-specific to produce the dark band or they have not co-immunoprecipitated HIF-1a (and the HIF-1a is at the size of the dark band). This needs to be explained or dropped. It’s the same for the HIF-2a blot on the right. The very dark band needs to be identified. There are no size markers either.

2) Supplementary figure 2- In the text the authors said that HIF-1a levels increased at 24hrs following transient knockdown. However, there is no evidence to show that HIF-1a was ever knocked down, so how can the authors say that HIF-1a was temporarily knocked down? It looks to me like they never knocked down the isoform. Cell death could easily be explained by toxicity from the transfection reagent.

These two supp figures and their discussion in the text must be addressed, corrected or removed from the paper as the current evidence seems to indicate the experiments themselves didn’t work.

Notes:

-The figures are can be complex and not always to understand. The paper is more likely to be understood by readers (and cited and used for further research) if the figures can be easily interpreted. Better legends and/or less abbreviations would help this. For example, figure 2 could have a legend indicating the name for the abbreviation for CCI rather than having to search through dense text to find out what CCI is.

-Pg 3, line 99- need to specify that the HIF-1a and HIF-2a knockdowns are in separate cell lines (not a double knockdown)

-Pg 5, lines 120 and 169: typos

-Figure 3B- western blot. This western does not have a solid band but rather transferred as two separate dots- I’m not sure how the authors could get quantitative data in the graph underneath it from this blot. There are some other westerns like this in the paper, and interpretation from incompletely transferred bands would not be an accurate representation.

-Pg 13, line 400- typo

Author Response

1.     Supplementary Figure 5. This figure and the interpretation are not acceptable in current form.  In the left blot, furthest left lane there is a very dark band at the bottom. My interpretation would be that this is the HIF-1a band but the authors claim that a higher (and very, very faint band) is actually the HIF-1a band. Either the antibody is VERY non-specific to produce the dark band or they have not co-immunopreciitated HIF-1a (and the HIF-1a is at the size of the dark band). This needs to be explained or dropped. It’s the same for the HIF-2a blot on the right. The very dark band needs to be identified. There are no size markers either.

We have organized better the presentation of the results in the new Supplementary Figure 5,  and we apologize for not have been explained them clearly before.  The Figure was corrected and the sisze markers appear now  indicated in the figure.

2.     Supplementary Figure 2. In the text the authors said that HIF-1a levels increased at 24 hrs following transient knockdown. However, there is no evidence to show that HIF-1a was ever knocked down, so how can the authors say that HIF-1a was temporarily knocked down? It looks to me like they never knocked down the isoform. Cell death could easily be explained by toxicity from the transfection reagent.

The Figure was corrected and re-organized appropriately. The results clearly show now that  HIF-1α and HIF-2α were efficiently silenced.

These two sup figures and their discussion in the text must be addressed, corrected or removed from the paper as the current evidence seems to indicate the experiments themselves didn’t work.

As we have explained above, both figures have been corrected taking into account all your suggestions.

Notes:

--The figures are can be complex and not always to understand. The paper is more likely to be understood by readers (and cited and used for further research) if the figures can be easily interpreted. Better legends and/or less abbreviations would help this. For example, figure 2 could have a legend indicating the name for the abbreviation for CCI rather than having to search through dense text to find out what CCI is.

Following your suggestions we have improved the figure legends avoiding the use of abbreviations.

---PG 3, line 99- need to specify that HIF-1a and HIF-2a knockdown are in separate cell lines (not a double knockdown)

We have corrected this replacing the sentence “we created a stable knockdown of HIF-1α and HIF-2α in….”   By the sentence: “we created a stable knockdown of HIF-1α or of  HIF-2α in….”

---Pg 5, lines 120 and 169: typos

The typos were corrected.

---Figure 3B:  western blot. This western does not have a solid band but rather as two separate dots. I’m not sure how the authors could get quantitative data in the graph underneath  it from this blot. There are some other westerns like this in the paper, and interpretation from incompletely transferred bands would not be an accurate representation.

We apologize for this. The representative western blot shown in Figure 3B was replaced by other showing completely transferred bands instead of separate dots.

---- Pg 13, line 400- typo

The typo was corrected.

We hope that all points raised by the Reviewers are satisfactorily answered and the manuscript be suitable for publication in Cancers. We also thank the Reviewers for their valuable comments and suggestions that allowed us to improve our work. 

Round 2

Reviewer 2 Report

All comments except supplementary figure 5 have been addressed. The darkest band shown in the far left column does not appear in the Co-IP. If the authors are claiming that the extremely faint band is HIF-1b in columns 2 and 3, then what is the very dark band under input? This band does not align with the very faint band in their co-IP columns. This would mean their antibody is more specific for something else or the dark band is HIF-1b and they have not co-immunoprecipitated any HIF-1b. The quality of the blot is poor for HIF-2a and I don't think it shows a specific difference, it looks like non-specific bands and dark smudge in the middle.

I actually think the authors could leave this data out, as it I don't think they have successfully co-IPed these proteins. PT-2385 is known to be specific to HIF-2a and I think it is acceptable to cite the work that showed it is specific without running these experiments (S5). Either way, this figure does not show me any data proving PT-2385 is specific for one isoform or another.

Author Response

We have followed your suggestion and Supplementary Figure 5 has now been removed from the manuscript.  We are grateful to you for your very valuable observations and  suggestions.

This manuscript is a resubmission of an earlier submission. The following is a list of the peer review reports and author responses from that submission.

Round 1

Reviewer 1 Report

Comments to authors

In this study Saint-Martin A et al. show that human colon cells lines express HIF1 and HIF2 factors already in normoxic conditions in line with their previous studies. Authors show this basal expression of HIF1a and HIF2a is functional because they control basal autophagy in colon tumor cells as well as their susceptibility to different chemotherapeutic agents and nutritional stress. These data are interesting. However authors should provide any evidence about molecular mechanisms underlying their findings.

Major comments

1.     Authors show that silencing of basal HIF1a and HIF2a expression elevates basal autophagy in human colon cancer cells. Authors should explore potential mechanisms involved in this observation. For example, taking into consideration the central role of mTORC1 in autophagy, authors should explore whether mTORC1 activity decreased in HIF1a and/or HIF2a-silenced cells.  Alternatively authors could explore whether HIF1a and/or HIF2 control the expression of those key players involved in autophagy shown in Supplemental Figure 1A.

2.     These data suggest that HIF1 and HIF2 act as inhibitors of basal autophagy in these colon tumor cells. Authors should discuss and cite a previous study showing opposite information in which HIF1a promotes cellular autophagy (PMID: 19273585).

3.     Authors propose that the increased autophagy is not involved in the reduced viability of HIF1 and HIF2-silenced cells (shown in Figure 2A) because HCQ - a presumed inhibitor of autophagy - do not restore viability of these HIF1/2-silenced cells. However authors should clarify why HCQ increase autophagy signal in Figure 1F considering that should presumably inhibit autophagy. Authors should clarify about this point. 

4.     Authors should also investigate potential mechanisms involved in the increased susceptibility of HIF2-silenced cells to nutritional stress plus mTORC1 inhibitor shown in Figure 3A. In line with comment 1, is mTORC1 repressed in a larger extent in HIF2a-silenced cells plus CCI-779 when compared with CCI-779 alone? 

5.     Finally, numerous studies have shown that HIF2a counteracts oxidative damage (for example, PMID: 19706526). Is increased apoptosis in HIF2-silenced cells a consequence of an exacerbated oxidative stress? In this line, authors should measure intracellular oxidative stress in control and HIF2-silenced cells under normal conditions as well as during nutritional stress plus CCI-779 treatment. In this line, authors should assess whether HIF2a-silenced cells show a reduced expression of antioxidant enzymes analyzed in the study referred above (PMID: 19706526). If HIF2-silenced cells show higher oxidative stress, authors should attempt to recover their viability using antioxidant treatment.   

Author Response

Reviewer #1:

3.Authors propose that the increased autophagy is not involved in the reduced viability of HIF1 and HIF2-silenced cells (shown in Figure 2A) because HCQ -a presumed inhibitor of autophagy inhibitor- do not restore viability of these HIF/2-silenced cells. However, authors should clarify why HCQ increase autophagy signal in Figure 1F considering that should presumably inhibit autophagy. Authors should clarify about this point.

We measured autophagy by quantifying autophagosome formation (the first step in the autophagy process) either visualizing the appearance of punctate structures (Figure 1E ) or by flow cytometry (Figure 1F). Because HCQ prevents the fusion of autophagosomes with lysosomes, the final step of autophagy, it provokes an accumulation of the autophagosomes at the cytoplasm. This is why in its presence the total amount of autophagosomes increase, but this means that the overall process has been inhibited. 

      1.Authors show that silencing of basal HIFia and HIF2a expression elevates basal autophagy in human colon cancer cells. Authors should explore potential mechanisms involved in this observation. For example, taking into consideration the central role of mTORC1 in autophagy, authors should explore whether mTORC1 activity decreased in HIF1a and /or HIF2a-silenced cells. Alternatively authors could explore whether HIF1a and/or HIF2a control the expression of those key players involved in autophagy shown in Supplemental Figure 1A.  2.These data suggest that HIF1 and HIF2 act as inhibitors of basal autophagy in these colon tumor cells. Author should discuss and cite a previous study showing opposite information in which HIFia promotes cellular autophagy (PMID: 19273585). 4.Authors should also investigate potential mechanisms involved in the increased susceptibility of HIF2-silenced cells to nutritional stress plus mTORC1 inhibitor shown in Figure 3A. In line with comment 1, is mTORC1 repressed in a larger extent in HIF2a-silenced cells plus CCI-779 when compared with CCI-779 alone?  5.Finally, numerous studies have shown that HIF2a counteracts oxidative damage (for example, PMID: 19706526). Is increased apoptosis in HIF2-silenced cells a consequence of an exacerbated oxidative stress? In this line, authors should measure intracellular oxidative stress in control and HIF2-silenced cells under normal conditions as well as during nutritional stress plus CCI-779 treatment. In this line, authors should assess whether HIF2a-silenced cells show a reduced expression of antioxidant enzymes analyzed in the study referred above (PMID: 19706526). If HIF-2-silenced cells show higher oxidative stress, authors should attempt to recover their viability using antioxidant treatment.  

The process of autophagy has been recognized as a major regulator of cellular viability under stressful conditions. This process can be triggered in cells by two mainly mechanisms:

a)     HIF-dependent autophagy: induced by HIF-1α /2α, which is mediated by BH3-only proteins BNIP3 and BNIPL3, and promote cell survival in normal or malignant cells.

b)     HIF-independent autophagy, positively regulated by AMPK and inhibited by mTORC1, which is induced by metabolic stress, nutrient depletion and oncogenic activation. Importantly, this type of autophagy in turn can positively modulate the HIF-dependent autophagy, because elevated oncogenic signaling in cancer cells induce HIF-α expression by inducing both their transcription and the stabilization and translation of HIF-α mRNAs [3, 6].

The human RKO and primary SW480 and its derivative metastatic SW620 cell lines used in our study  are representative of a range of BRAF-driven and KRAS-driven colorectal genotypes, respectively [15]. We must take into account that in B-RAF-driven and RAS-driven cancer cells HIF-independent autophagy is particularly robust. Consistent with this, these cells are considered as “addicted to autophagy” [17, 30-33]. Why?

Cancer cells depend on ATP-demanding anabolic processes activated by mTORC1 to produce building blocks for cell proliferation such as proteins, lipids and nucleotides. Oncogenic Ras impairs acetyl-CoA production through several mechanisms leaving cells dependent on autophagy to provide substrates for biosynthetic processes.  mTORC1 functions as a master regulator of cell growth, metabolism and autophagy, contributing to many aspects of cancer cell survival. However, oncogenic hyperactivation of mTOR would inhibit autophagy and would induce apoptosis as a result of Akt inhibition via negative feedback loop. In addition, increased protein synthesis in mTORC1-activated cells induces ER stress and triggers the unfolded protein response.  Therefore, mTORC needs to be counterbalanced in cancer cells to obtain at the same time the benefits of its activity, but also the benefits of activating autophagy avoiding induction of cell death.

On the other hand, the high metabolic demand and insufficient vascularization provoke not only ER stress, but hypoxia and oxidative stress, all of them inducing stress responses  and affecting autophagy to restore cellular homeostasis. Indeed, oxidative stress, hypoxia, and ER stress are closely intertwined and cancer cells develop protective mechanisms to cope with them promoting cell survival. Examples of such mechanisms to adapt to chronic stress are the metabolic reprogramming (the Warburg effect), antioxidant protein synthesis induction, angiogenesis induction, and autophagy addiction.

Our results are not in conflict with other papers we cite because the following reasons: i) other authors have also reported that the loss of one HIF-α subunit is compensated by upregulating the remaining HIF-α isoform, either in cancer cells [12,34,35] or in non-malignant systems such as cartilage [13]; ii) importantly, other authors have also found that the knockdown of HIF-2a results in a great increase in autophagy levels [12, 13], suggesting that lacking one HIF-α isoform up-regulate the other and thereby elicit survival advantages by dysregulating autophagy.

We apologize for not having being clear before in explaining our apparently contradictory results obtained as a result of HIFs-silencing expression. We have now clearly explained in the Results section (Figure 1G and Supplementary Figure S2) and at the discussion.

You are right pointing out that if both HIF isoforms favors autophagy in cells as survival mechanism under stress, it would be expected that the knockdown of each one of them in colon cancer cells would negatively affected autophagy levels. However, what we obtained was the opposite effect:  the knockdown of either HIF-1a or HIF-2a produced a paradoxical increase in autophagy levels.  Importantly, at the same time we also obtained here that the depleted expression of one HIF-α subunit was compensated by upregulating the remaining HIF-α isoform. Since it has been reported that both HIF-1α and HIF-2α appear to be evenly  matched in their capacity to induce autophagy [31], we investigated whether this paradoxical incresase in autophagy induction may be the result of the compensatory expression we previously observed when one HIF-α subunit is depleted. Because we could not get the simultaneous knockdown of HIF-1/2α factors (shown in Supplementary Figure S2), we then tried to mimic the effects of the double HIF-1α /2α silencing combining with the use of PT-2385, a ligand that blocks specifically the heterodimerization of HIF-2α with HIF-1b needed to activate the transcription of genes regulated by HIF-2α, with stable HIF-1α -silenced cells. The autophagy levels quantified by flow cytometry presented in the new Figure 1G show how the incubation of stable HIF-1α-silenced cells with the HIF-2α-specific antagonist reverted (indicated with a red arrow) the increase in autophagy levels produced as a result of HIF-1α –silencing, which displayed a compensatory HIF-2α expression (Figure 1D). However, because in this type of cancer cells, HIF-independent autophagy is particularly robust, we can not rule out that the stressful condition imposed by the depletion of one HIF-α isoform could also activate the HIF-independent autophagic pathway and thus, the transcription and/or translation of HIFs induced by oncogenic activity, contributing to autophagy induction. In support of this, it is interestingly to note in Supplementary Figure S2 that when we tried to transiently knockdown HIF-1α in stable silenced HIF-2α SW480 cells, surprisingly, the stable depleted HIF-2α expression was recuperated after 24 h post-transfection with the sh-HIF-1α plasmid.

Finally, with respect to your question 4 (is mTORC1 repressed in a larger extent in HIF2α-silenced cells plus CCI-779 when compared with CCI-779 alone?: we have studied how mTORC1 and mTORC2 participate in modulating autophagy and we have found that although mTORC1 can be modulated in cancer cells to avoid its hyperactivation, it is never repressed as a result of HIF-α silencing. However, we have found that CCI-779 treatment of cells in any condition always potently inhibits mTORC1. We have also studied how HIF-α isoforms counteract oxidative damage in colon cancer cells.   However, because we consider that with the new experiments performed, and the appropriate explanations added in the manuscript, we have clarified the main concern expressed in your comments regarding the conflicting results.  This clarification was essential to understand the manuscript, but we have taken into account that the main contribution of this study was the demonstration that HIF-2α plays a crucial role in promoting drug resistance and cell survival of patient’s-derived colon cancer cells.  How mTORC, hypoxia and oxidative stress participate in autophagy regulation is complex and as mentioned before, they are closely intertwined in cancer cells promoting cell survival. Thus, in view of this, and in the fact that we have many data about this that cannot be added at the manuscript since we have reached the maximum allowed limit (6 Figures with several panel each one plus 6 supplementary Figures), we have not included the additional data of mTORC and oxidative stress in this manuscript.

We thank you for your valuable comments and suggestions that allowed us to improve our work. We hope that our manuscript be now suitable for publication in Cancers.

Reviewer 2 Report

In the introduction, the authors cite work indicating that high levels of HIF-αs are associated with increased autophagy in multiple cancers.

The authors found that the colon cancer cells have increased basal autophagy and elevated HIFs as compared to healthy cells. The authors also cite a paper (10) that found that knockdown of HIF-1α decreases autophagy.

However, when the authors silence HIF-1α or HIF-2α, autophagy increases rather than decreases as would be expected based on previous results (?). If autophagy is a protective and survival mechanism and HIF-1α or HIF-2α knockdown increases it, I’m confused by the authors’ conclusions. They seem to be implying that HIFs are leading to increased autophagy, yet the first experiment where a single isoform is knocked out leads to increased autophagy rather than decreased. Further, their attempts at a double HIF-1α/HIF-2α knockdown failed (killing cells).

A quick easy experiment to resolve this, would be to do a transient double knockdown of both HIF-α isoforms using siRNAs. We have had success in simultaneously knocking down both isoforms with commercially available HIF-1α and HIF-2α siRNAs without cell death issues over short periods (a couple days) rather than stable cell lines. This would enable assessing basal autophagy levels driven by both individual isoforms and when BOTH isoforms are knocked down. A transient double knockdown would be expected to decrease autophagy if HIF-α total is driving it.

It is otherwise hard move past this confusing data and interpretation.

If the authors could clarify this section, with either the double knockdown data or if I have mis-interpreted this, then rewriting to clarify this, I am happy to finish reviewing the rest of the manuscript which looks interesting.

Notes for clarification: Do the authors think knockdown of a single HIF-α not only increases the other isoform to compensatory levels, but increases it to even higher total levels of a single isoform as compared to cells with both isoforms? (ie control cells = 1x HIF-1α and 1x HIF-2α. Then a HIF-1α knockdown results in 0x HIF-1α and 4x HIF-2α?). Are they arguing that there is a ratio imbalance that affects autophagic induction? The double knockdown is essential to verifying that this is the case.

If the double knockdown completely fails even transiently, then an EMSA or ELISA or qPCR data is needed to quantify levels of a particular HIF-α isoform and possibly downstream HIF gene expression activity. This is particularly because the results otherwise conflict with the papers they cite (knockdown of a HIF-α increases autophagy when the autophagy is supposed to HIF-α driven).

Clarifying this section is essential to understanding the manuscript.

Author Response

Reviewer #2:

In the introduction, the authors cite work indicating that high levels of HIF-as are associated with increased autophagy in multiple cancers. The authors found that the colon cancer cells have increased basal autophagy and elevated HIFs as compared to healthy cells. The authors also cite a paper (10) that found that knockdown of HIF-1a decreases autophagy. ]However, when the authors silence HIF-1a or HIF-2a, autophagy increases rather than decreases as would be expected based on previous results (?). If autophagy is a protective and survival mechanism and  HIF-1a or HIF-2a knockdown increases it, I’m confused by the author’s conclusions. They seem to be  implying that HIFs are leading to increased autophagy, yet the first experiment where a single isoform is knocked out leads to increased autophagy rather than decreased. Further, their attempts at a double HIF-1a/HIF-2a knockdown failed (killing cells). 

A quick easy experiment to resolve this, would be to do a transient double knockdown of both HIF-a isoforms using siRNAs. We have had success in simultaneously knocking down both isoforms with commercially available HIF-1a and HIF-2a siRNAs without cell death issues over short periods (a couple days) rather than stable cell lines. This would enable assessing basal autophagy levels driven by both individual isoforms and when BOTH isoforms are knocked down. A transient double knockdown would be expected to decrease autophagy if HIF-a total is driving it.

It is otherwise hard to move past this confusing data and interpretation. If the authors could clarify this section, with either the double knockdown data or if I have misinterpreted this, then rewriting to clarify this, I am happy to finish reviewing the rest of the manuscript which looks interesting.

Notes for clarification: do the authors think knockdown of a single HIF-a not only increases the other isoform to compensatory levels, but increases it to even higher total levels of a single isoform as compared to cells with both isoforms? As they arguing that there is a ratio imbalance that affects autophagic induction? The double knockdown is essential to verifying that this is the case.

If the double knockdown completely fails even transiently then an EMSA or ELISA or qPCR data is needed  to quantify levels of a particular HIF-a isoform and possible downstream HIF gene expression activity. This is particularly because the results otherwise conflict with the papers they cite (knockdown of a HIF-a increases autophagy when autophagy is supposed to HIF-a driven).  Clarifying this section is essential to understanding the manuscript.

The process of autophagy has been recognized as a major regulator of cellular viability under stressful conditions. This process can be triggered in cells by two mainly mechanisms:

a)     HIF-dependent autophagy: induced by HIF-1α /2α, which is mediated by BH3-only proteins BNIP3 and BNIPL3,  and promote cell survival in normal or malignant cells.

b)     HIF-independent autophagy, positively regulated by AMPK and inhibited by mTORC1, which is induced by metabolic stresses, nutrient depletion and oncogenic activation. Importantly, this type of autophagy in turn can positively modulate the HIF-dependent autophagy, because elevated oncogenic signaling in cancer cells induce HIF-α expression by inducing both their transcription and the stabilization and translation of HIF-α mRNAs [3, 6].

The human RKO and primary SW480 and its derivative metastatic SW620 cell lines used in our study  are representative of a range of BRAF-driven and KRAS-driven colorectal genotypes, respectively [15]. We must take into account that in B-RAF-driven and RAS-driven cancer cells HIF-independent autophagy is particularly robust. Consistent with this, these cells are considered as “addicted to autophagy” [17, 30-33]. Why?

Cancer cells depend on ATP-demanding anabolic processes activated by mTORC1 to produce building blocks for cell proliferation such as proteins, lipids and nucleotides. Oncogenic Ras impairs acetyl-CoA production through several mechanisms leaving cells dependent on autophagy to provide substrates for biosynthetic processes.  mTORC1 functions as a master regulator of cell growth, metabolism and autophagy, contributing to many aspects of cancer cell survival. However, oncogenic hyperactivation of mTOR would inhibit autophagy and would induce apoptosis as a result of Akt inhibition via negative feedback loop. In addition, increased protein synthesis in mTORC1-activated cells induces ER stress and triggers the unfolded protein response.  Therefore, mTORC needs to be counterbalanced in cancer cells to obtain at the same time the benefits of its activity, but also the benefits of activating autophagy avoiding induction of cell death.  On the other hand, the high metabolic demand and insufficient vascularization provoke not only ER stress, but hypoxia and oxidative stress, all of them inducing stress responses  and affecting autophagy to restore cellular homeostasis. Indeed, oxidative stress, hypoxia, and ER stress are closely intertwined and cancer cells develop protective mechanisms to cope with them promoting cell survival. Examples of such mechanisms to adapt to chronic stress are the metabolic reprogramming (the Warburg effect), antioxidant protein synthesis induction, angiogenesis induction, and autophagy addiction.

Our results are not in conflict with other papers we cite because the following reasons: i) other authors have also reported that the loss of one HIF-α subunit is compensated by upregulating the remaining HIF-α isoform, either in cancer cells [12,34,35] or in non-malignant systems such as cartilage [13]; ii) importantly, other authors have also found that the knockdown of HIF-2α  results in a great increase in autophagy levels [12, 13], suggesting that lacking one HIF-α isoform up-regulate the other and thereby elicit survival advantages by dysregulating autophagy.

We apologize for not having being clear before in explaining our apparently contradictory results obtained as a result of HIFs-silencing expression. We have now clearly explained in the Results section (Figure 1G and Supplementary Figure S2) and at the discussion. You are right pointing out that if both HIF isoforms favors autophagy in cells as survival mechanism under stress, it would be expected that the knockdown of each one of them in colon cancer cells would negatively affected autophagy levels. However, what we obtained was the opposite effect:  the knockdown of either HIF-1α or HIF-2α produced a paradoxical increase in autophagy levels.  Importantly, at the same time we also obtained here that the depleted expression of one HIF-α subunit was compensated by upregulating the remaining HIF-α isoform. Since it has been reported that both HIF-1α and HIF-2α appear to be evenly  matched in their capacity to induce autophagy [31], we investigated whether this paradoxical incresase in autophagy induction may be the result of the compensatory expression we previously observed when one HIF-α subunit is depleted. Because we could not get the simultaneous knockdown of HIF-1/2α factors (shown in Supplementary Figure S2), we then tried to mimic the effects of the double HIF-1α /2α silencing combining with the use of PT-2385, a ligand that blocks specifically the heterodimerization of HIF-2α with HIF-1β needed to activate the transcription of genes regulated by HIF-2α, with stable HIF-1α -silenced cells. The autophagy levels quantified by flow cytometry presented in the new Figure 1G show how the incubation of stable HIF-1α-silenced cells with the HIF-2α-specific antagonist reverted (indicated with a red arrow) the increase in autophagy levels produced as a result of HIF-1α silencing, which displayed a compensatory HIF-2α expression (Figure 1D). However, because in this type of cancer cells, HIF-independent autophagy is particularly robust, we can not rule out that the stressful condition imposed by the depletion of one HIF-α isoform could also activate the HIF-independent autophagic pathway and thus, the transcription and/or translation of HIFs induced by oncogenic activity, contributing to autophagy induction. In support of this, it is interestingly to note in Supplementary Figure S2 that when we tried to transiently knockdown HIF-1α in stable silenced HIF-2α SW480 cells, surprisingly, the stable depleted HIF-2α expression was recuperated after 24 h post-transfection with the sh-HIF-1α plasmid.

We thank you for your valuable comments and suggestions that allowed us to improve our work. We hope that our manuscript be now suitable for publication in Cancers.